# Low-temperature nucleation anomaly in silicate glasses shown to be artifact in a 5BaO·8SiO$_2$ glass

Xinsheng Xia[1], D. C. Van Hoesen[2], Matthew E. McKenzie[3], Randall E. Youngman [3] & K. F. Kelton [1,2✉]

For over 40 years, measurements of the nucleation rates in a large number of silicate glasses have indicated a breakdown in the Classical Nucleation Theory at temperatures below that of the peak nucleation rate. The data show that instead of steadily decreasing with decreasing temperature, the work of critical cluster formation enters a plateau and even starts to increase. Many explanations have been offered to explain this anomaly, but none have provided a satisfactory answer. We present an experimental approach to demonstrate explicitly for the example of a 5BaO • 8SiO$_2$ glass that the anomaly is not a real phenomenon, but instead an artifact arising from an insufficient heating time at low temperatures. Heating times much longer than previously used at a temperature 50 K below the peak nucleation rate temperature give results that are consistent with the predictions of the Classical Nucleation Theory. These results raise the question of whether the claimed anomaly is also an artifact in other glasses.

[1] Institute of Materials Science and Engineering, Washington University, St. Louis 63130 MO, USA. [2] Department of Physics, Washington University, St. Louis 63130 MO, USA. [3] Science and Technology Division, Corning Incorporated, Corning 14831 NY, USA. ✉email: kfk@wustl.edu

The development of more quantitative models for nucleation in silicate glasses is critical for accelerating the production of new glasses and glass ceramics with tailored microstructures[1,2]. For the commonly used Classical Nucleation Theory (CNT), the competition between the thermodynamic-driving free energy and the kinetics as a function of temperature gives a maximum nucleation rate as a function of temperature[3]. However, experimental studies made over the past four decades in many silicate glasses have shown that the measured time-dependent nucleation rates at temperatures below the temperature of the maximum nucleation rate contradict the predictions of the CNT[4–8]. The critical work of cluster formation (nucleation barrier), $W^*$, should decrease monotonically with decreasing temperature due to its relation to the thermodynamic driving free energy and interfacial free energy[5,8]. However, as shown in Fig. 1, the experimental results (scaled to $k_BT$, where $k_B$ is Boltzmann's constant and $T$ is the temperature) from the literature indicate that at low temperatures the nucleation barrier levels off or even increases with decreasing temperature[7–16]. There have been several attempts to explain this low-temperature anomaly[7–9,17]. Abyzov et al.[7] showed that the anomaly cannot be explained as an elastic strain energy effect. Fokin et al.[8] argued that it could be explained by adjusting the volume of the structural unit at different nucleation temperatures. Gupta et al.[17] suggested that the size of the cooperatively rearranging regions could be the reason for the low-temperature nucleation anomaly. Abyzov et al.[9] proposed spatial heterogeneities, where nucleation proceeds only in liquid-like regions. Already in some previous studies (Zanotto et al.[18] and Greer et al.[19]), the possibility was raised that the nucleation anomaly might be an artifact, but without providing conclusive evidence. A series of previously published nucleation data sets were recently re-analyzed by Cassar et al.[20], focusing on data near the peak nucleation temperature. They concluded that not all data points could be taken with equal confidence, finding variations even across data sets for the same type of glass. From this, they cast doubt on the widely studied nucleation anomaly. Partially motivated by the conclusions of Cassar et al.[20] and by those from other data analyses (such as Gupta et al.[17]), we

concluded that the anomaly might be an artifact resulting from insufficient heating time at the low nucleation temperatures.

Here we show that the anomaly previously reported in a $5BaO \cdot 8SiO_2$ glass[10] was indeed an experimental artifact. This was demonstrated by using a suitably designed experimental procedure and tracking the nucleation process over extensively long periods of time. The time-dependent nucleation rate was measured in the $5BaO \cdot 8SiO_2$ glasses that were held at a nucleation temperature of 948 K, which is 50 K below the temperature of the maximum nucleation rate, for up to 115 days. This time is much longer than any used in earlier studies of silicate glasses[10–16,18,21,22]. Previous studies of silicate glasses have argued that the critical work of cluster formation plateaus or increases with decreasing temperatures below the peak nucleation temperature (Fig. 1). The new experimental data for $5BaO \cdot 8SiO_2$ instead show that the critical work of cluster formation monotonically decreases with decreasing temperature, following the trend expected from the Classical Nucleation Theory. The data therefore confirm the suggestion by Cassar et al.[20] that the nucleation anomaly at low temperatures is not a real phenomenon in all silicate glasses, but is rather an experimental artifact, at least in this $5BaO \cdot 8SiO_2$ glass, due to the short nucleation times used in earlier studies.

## Results

**Nucleation rate and induction time.** The approach used to measure the nucleation rate is discussed in the "Methods" section; the results are discussed here. Figure 2 shows the measured number of nuclei per unit volume, $N_v$, as a function of nucleation time at 948 K, together with data measured for this same glass earlier[10]. Initially, $N_v$ increases nonlinearly with time, a phenomenon widely recognized for nucleation in melt-quenched glasses as due to the evolving cluster population as a function of cluster size; $N_v$ eventually becomes linear with time, indicating that steady-state has been reached. The steady-state nucleation rate ($I^{st}$) and the induction time ($\theta_{n^*(T_G)}$) are obtained from the slope and intercept with the time axis, respectively, of the linear portion of the curve[3]. The measured values are $I^{st} = 400 \pm 20$ mm$^{-3}$ s$^{-1}$ and $\theta_{n^*(T_G)} = 40,000 \pm 3000$ minutes. These values are listed in Table 1, together with our previous results[10]. With

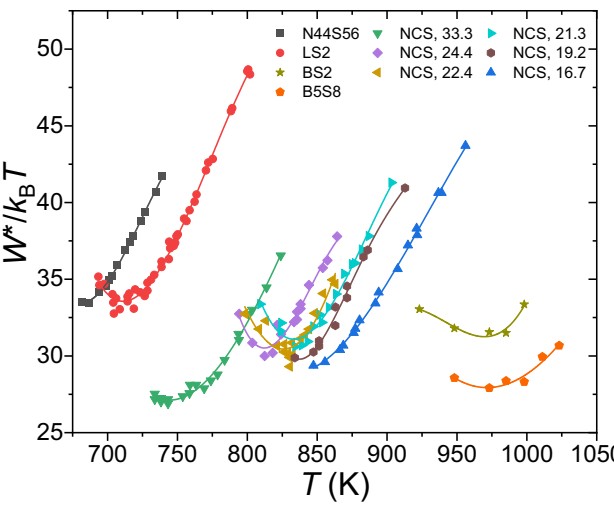

**Fig. 1 The scaled nucleation barrier as a function of temperature for silicate glasses.** These data are from the literature and for the following glasses: $44Na_2O \cdot 56SiO_2$ (N44S56)[8,16], $Li_2O \cdot 2SiO_2$ (LS2)[8,11], $BaO \cdot 2SiO_2$ (BS2)[10], $5BaO \cdot 8SiO_2$ (B5S8)[10], and $xNa_2O \cdot (50-x)CaO \cdot 50SiO_2$ (NCS) where the x-values are 33.3[8,15], 24.4[7,13], 22.4[7,13], 21.3[7,13], 19.2[7,13], and 16.7[8,12–14], respectively. The solid lines serve as guides to the eye. (Reproduced from refs. [7,8,10] with permission from Elsevier.)

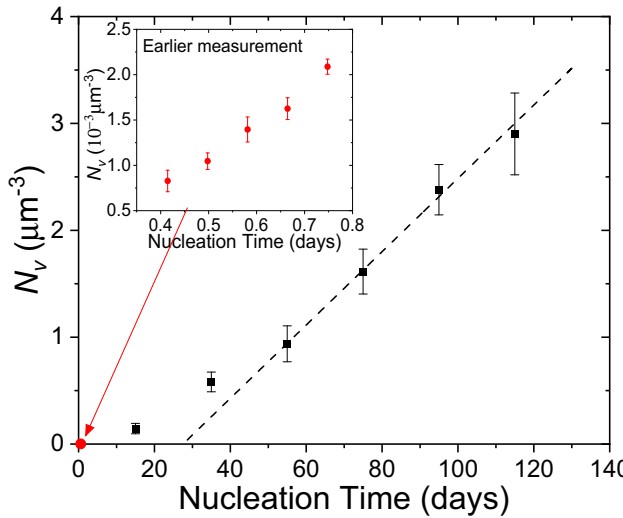

**Fig. 2 The number of nuclei per unit volume as a function of nucleation time at 948 K for the $5BaO \cdot 8SiO_2$ glasses.** The black points are the new data obtained here and the red points are the data from an earlier study[10] (see inset, reproduced from ref. [10] with permission from Elsevier). The dashed lines show the linear fits in the steady-state range. (The error bars indicate the SD.)

the significantly longer nucleation time, the new values of $I^{st}$ and $\theta_{n^*(T_G)}$ are 7 times and 111 times, respectively, larger than the values obtained in the previous study[10].

**Interfacial free energy and critical work of cluster formation**. The methods used to obtain the interfacial free energy $\sigma$, the critical work of cluster formation $W^*$, and the induction time for

**Table 1 Steady-state rates and induction times for nucleation in 5BaO·8SiO₂ glasses.**

| Temperature, $T$ (K) | Steady-state nucleation rate, $I^{st}$ (mm$^{-3}$ s$^{-1}$) | Induction time, $\theta_{n^*(T_G)}$ (minutes) |
|---|---|---|
| 948 (this measurement) | 400 ± 20 | 40,000 ± 3000 |
| 948* | 48 ± 3 | 354 ± 41 |
| 973* | 746 ± 72 | 45 ± 4 |
| 985* | 1345 ± 25 | 16.1 ± 0.4 |
| 998* | 3135 ± 54 | 7.4 ± 0.3 |
| 1011* | 2599 ± 127 | 1.8 ± 0.2 |
| 1023* | 2035 ± 28 | 1.1 ± 0.1 |
| 1048* | 669 ± 53 | Not determined |

948 K (this measurement) is the measurement here using 1073 K as the growth temperature. All the data labeled with * are from our previous study[10] (reproduced from ref. [10] with permission from Elsevier), which used 1119 K as the growth temperature. The value and SE were determined from the linear fit in the $N_v$ vs. nucleation time plots using the instrumental weighting in Origin software.

the critical size at the nucleation temperature $\theta_{n^*(T_N)}$ from the nucleation data are discussed in the Supplementary Method 1 in the Supplementary Information file. The values for $I^{st}$ and $\theta_{n^*(T_G)}$ at 948 K from this study were combined with values obtained at temperatures at or above the temperature for the maximum steady-state nucleation rate, whose $I^{st}$ and $\theta_{n^*(T_G)}$ are known, previously reported by Xia et al.[10]. The measured induction time corresponds to that for the critical size at the growth temperature, $\theta_{n^*(T_G)}$. To compare with predictions of CNT, the induction time for the critical size at the nucleation temperature, $\theta_{n^*(T_N)}$, is required. This was computed from $\theta_{n^*(T_G)}$ following a method discussed earlier.[10] The Turnbull approximation from the enthalpy of fusion and the liquidus temperature[10,23] was used to calculate the driving free energy as a function of temperature, $|\Delta g_v|$, assuming one unit of 5BaO · 8SiO₂ (Fig. 3a). The calculated interfacial free energy, $\sigma$, is shown in Fig. 3b (the details of how $\sigma$ was calculated are given in the Supplementary Method 1 in the Supplementary Information file), along with the values obtained previously[10]. The previous results showed that although at high temperature $\sigma$ decreases linearly with decreasing temperature, this changed to an increasing $\sigma$ with decreasing temperature for temperatures below the temperature for maximum nucleation rate (998 K). The new measurements obtained here show that $\sigma$ monotonically decreases with decreasing temperature over the whole temperature range, consistent with the predictions of the Diffuse Interface Theory of nucleation[24–27]. In addition, unlike the previous results[10] (Fig. 3c), $W^*/k_B T$ decreases over the entire

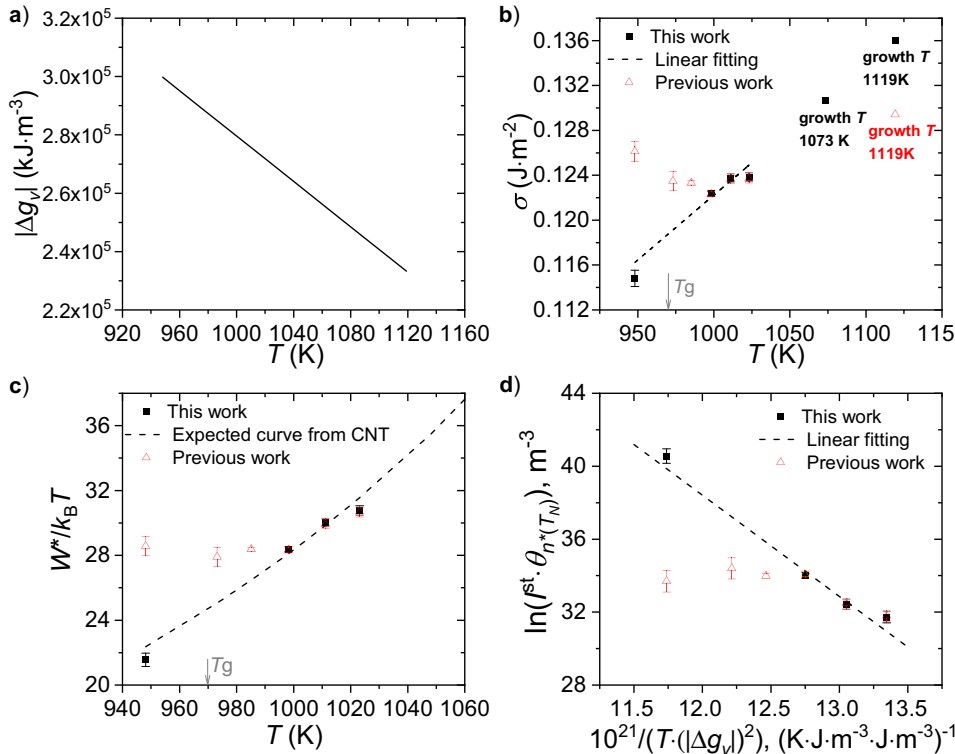

**Fig. 3 The values of nucleation parameters obtained from this study and from the previous study[10].** This study uses the longer nucleation time at 948 K, and the previous study[10] used shorter-time nucleation data at low temperatures for 5BaO•8SiO₂ glasses. **a** The driving free energy used as a function of temperature. **b** The interfacial free energy obtained as a function of temperature. **c** The scaled nucleation barrier obtained as a function of temperature. **d** The natural logarithm of the product of the steady-state nucleation rate and the induction time for the critical size at the nucleation temperature, as a function of the reciprocal of the product between temperature and the square of driving free energy. The errors were calculated using the 95% confidence intervals of the steady-state nucleation rate and the induction time. The red symbols represent the values obtained in the previous study[10] (reproduced from ref. [10] with permission from Elsevier). $T_g$ is the glass transition temperature.

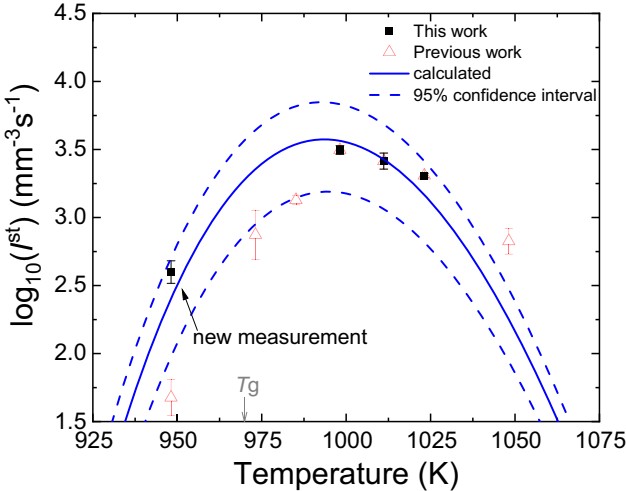

**Fig. 4 A comparison between the calculated steady-state nucleation rate from the Classical Nucleation Theory and the measured data.** The solid blue curve is the calculated rate from theory. The red symbols represent the values obtained in the previous study[10]. The dashed lines are the 95% confidence limits for the calculated curve. The error bars are the 95% confidence intervals of the measured data. $T_g$ is the glass transition temperature.

temperature range, rather than decreasing with decreasing temperature only when the temperature is higher than the peak nucleation temperature (998 K) but plateauing at lower temperatures. The new results follow the trend expected from CNT. Finally, CNT predicts that a plot of $\ln(I^{st}\theta_{n^*(T_N)})$ as a function of $1/(T|\Delta g_v|^2)$ should be linear[4] when $\sigma$ is a constant or the relative change in $\sigma$ as a function of temperature is smaller than the relative change in $|\Delta g_v|$ as a function of temperature. As shown in Fig. 3d, this is true if the new data are used, as opposed with the previous results showing a significant departure from the straight line behavior[10].

**Comparison between measured and theoretical nucleation rates.** The nucleation rate as a function of temperature was calculated assuming CNT and using the values for $|\Delta g_v|$ shown in Fig. 3a and $\sigma$ given by the dashed line in Fig. 3b, and assuming the Kashchiev expression[28] (shown in the Supplementary Method 2 in the Supplementary Information file) to calculate the diffusion coefficient from the induction time for the critical size at the nucleation temperature. The result is shown by the solid line in Fig. 4; the 95% confidence bounds are indicated by the dashed lines. Except for the data point at the highest temperature (shown in red), the high-temperature data and the new low-temperature data point (shown in black) agree reasonably well with the calculated nucleation rates. Importantly, the two data points at 973 K and 985 K fall below or close to the lower limit of the 95% confidence bounds, indicating that they have not yet achieved the steady-state value. The data point at 1048 K falls outside of the higher limit of the confidence bounds. However, this is likely to be an artifact of the fit. The induction time was not measurable at this temperature; instead, it was estimated from the data at 998 K, 1011 K, and 1023 K.

**Discussion**

In summary, the time-dependent nucleation rate was measured in $5BaO \cdot 8SiO_2$ glasses at a temperature that was 50 K below the peak nucleation rate temperature. Earlier measurements of the

steady-state nucleation rate in this glass[10] showed an anomalous behavior at these low temperatures, which was consistent with what has been reported in many other silicate glasses[4–8]. For the new measurements reported here, the glasses were given a much longer nucleation treatment than was used in all previous measurements of silicate glasses[10–16,18,21,22]. These new data do not show a low-temperature anomaly. In contradiction to previous results, the interfacial free energy decreases with decreasing temperature over the whole measurement temperature range, consistent with predictions of the diffuse interface theory of nucleation[24–26]. Also, following the trend predicted by the Classical Nucleation Theory, the critical work of cluster formation monotonically decreases with decreasing temperature instead of plateauing or increasing with decreasing temperature for temperatures below the peak nucleation temperature, which the earlier studies showed. These results demonstrate that the anomaly is not a real phenomenon, but is an experimental artifact (at least in the $5BaO \cdot 8SiO_2$ glass studied here) due to insufficient nucleation treatment times at low temperatures in previous studies. Based on this result and given the practical importance of knowing the nucleation rate as a function of temperature, the low-temperature data in other silicate glasses should be re-measured, as they are possibly incorrect and the anomaly similarly not real.

**Methods**

**Choice of materials and glass preparation.** A barium-silicate glass was chosen for this study, as it has larger nucleation rates than silicate glasses such as $Li_2O \cdot 2SiO_2$ or $Na_2O \cdot 2CaO \cdot 3SiO_2$, thus requiring less time to obtain a significant number of nuclei. The crystals in the $5BaO \cdot 8SiO_2$ glasses are also spherical, making it easier to accurately measure the nuclei density than in the $BaO \cdot 2SiO_2$ glass, e.g., where the crystals have irregular shapes[10]. The $5BaO \cdot 8SiO_2$ glasses were prepared by Corning Incorporated using the melting and quenching procedures discussed by Xia et al.[10]. The source materials were barium carbonate and silica. In platinum crucibles, 2500 g of the mixed source materials were melted at 1873 K for 6 h, quenched, broken, re-melted at 1773 to 1873 K for 6 h, and quenched on a stainless steel table or roller quenched to form glasses. The composition of the prepared bulk glasses was measured by inductively coupled plasma–optical emission spectroscopy to be BaO (38.73 mol%), $SiO_2$ (61.21 mol %), SrO (0.04 mol%), $Fe_2O_3$ (0.01 mol%), and $Al_2O_3$ (<0.01 mol%). As reported earlier[10], the measured glass transition temperature for the $5BaO \cdot 8SiO_2$ glass is 970 K. Prior to the heat treatments, the bulk $5BaO \cdot 8SiO_2$ glasses were cut into plates having an area of ~3.8 mm × 3.0 mm and a thickness of 0.98 ± 0.07 mm (average ± SD).

**Heat treatments.** The time-dependent nucleation rate was measured using the two-step heating method[21,29]. Samples were first heated at a temperature where the nucleation rate is large, but the growth velocity is small. These nuclei were then grown to observable size by heating at a temperature where the growth velocity is larger than that at nucleation temperature but the nucleation rate is small. During the nucleation treatment, the samples were heated together in a container (a 5 mL Coors high alumina combustion boat, Sigma Aldrich) in a Lindberg tube furnace at 948 ± 2 K (the temperature range of the center of the furnace). To mitigate possible diffusion between the samples and the container, an additional spacer of $5BaO \cdot 8SiO_2$ glass (~1 cm thick) was placed between the two. The spacer was replaced with a new one every 25 or 30 days. When each target heating time was reached, the collection of samples and container were taken out of the furnace, air quenched onto a metal plate to room temperature, and one sample was randomly removed. The remaining samples were then reinserted into the furnace and positioned close to the center of the 948 ± 2 K temperature range in the furnace. Samples were nucleated for 15, 35, 55, 75, 95, and 115 days. The nuclei density in these samples, which had been held at the nucleation temperature for a much longer time than in previous studies[10–16,18,21,22], was so large that due to crystal impingement they could not be grown to sizes that could be observed in optical microscopy. Instead, a growth treatment was selected that produced crystals with diameters smaller than 1 μm; the nuclei density was then measured in a scanning electron microscope (SEM). After the nucleation treatment, each sample was placed inside a 5 mL Coors high alumina combustion boat (Sigma Aldrich) and inserted into a Lindberg Blue M three-zone tube furnace that had been equilibrated at 1073 K. Eight minutes after insertion, the sample and the boat were removed from the furnace and air quenched onto a metal plate. The number of the new nuclei formed during the growth treatment was negligible compared with the number of nuclei created during the nucleation treatment.

**Polishing, etching, cleaning, imaging, and image analysis methods.** After the nucleation and growth heat treatments, the samples were polished, etched, and cleaned following the similar procedures used previously[10]. At least 250 μm thickness of the sample surfaces were removed during polishing, using 400-, 600-, 800-grit silicon carbide papers and a 0.5 μm ceria suspension (Allied High Tech Products, Inc.) with running water. After etching in a 0.2 HCl 0.5 HF (vol%) etchant solution for 10 s, the samples were cleaned with deionized water. Then the samples were further ultrasonically cleaned in acetone, ultrasonically cleaned in deionized water, and finally dried on tissue paper. The nuclei density was determined by imaging the sample surface using a Thermofisher Quattro S Environmental SEM with a 10 kV accelerating voltage, 30 Pa chamber pressure, and a low-vacuum detector operating in the secondary electron mode. At least 11 SEM images were taken from each sample. Typical SEM images showing spherical crystals are shown in Supplementary Figs. 1–6 in the Supplementary Information file. For each image, the number of crystals per area, $N_s$, and the average of the reciprocal diameters, $\bar{Y}$, were measured. The number of crystals per unit volume, $N_v$, was determined using[30,31]

$$N_v = \frac{2}{\pi} N_s \bar{Y}. \tag{1}$$

For each sample, the SD for $N_v$ was calculated from multiple images. The microscopy resolution limit-related correction for a monodispersed system[32] and the density of nuclei in the as-quenched glass were used to further correct $N_v$.

## Data availability

All of the original SEM images are available from the corresponding author upon request. Typical original SEM images for each sample are included in the Supplementary Information file. All data generated or analyzed during this study are included in this published article.

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

## Acknowledgements

We thank E. Zanotto and D. Cassar for pointing out their statistical analysis and for very useful discussions. We also thank M. Sellers, R. Chang, A. Gangopadhyay, and P. Gibbons for useful discussions. We acknowledge the Institute of Materials Science and Engineering at Washington University in St. Louis for the use of the Thermofisher Quattro S Environmental SEM. This work is financially supported by NSF GOALI grant DMR 17-20296 and by Corning Incorporated.

## Author contributions

K.F.K. conceived of the idea for this study. X.X. detailed the experiment plan, conducted the nucleation measurements, and analyzed the data. D.C.V. assisted with the calibration of furnace, the etching process, and the data analysis. K.F.K., X.X., D.C.V., M.E.M., and R.E.Y. engaged in useful discussions during the ongoing study and contributed to the writing and editing of the paper.

## Competing interests

The authors declare no competing interests.
