## [Peer Review File · Nature Communications]

REVIEWERS' COMMENTS

Reviewer #1 (Remarks to the Author):

Response to the authors

Second round by reviewer 1(following the Nature Materials sequence)

We thank the referees for a careful and critical review of our manuscript. They have
made several important points that we have addressed in the revised manuscript and
in this response. Before addressing each of the referee's comments in turn, we
would like to indicate the major revisions made in the revised manuscript.

1. To make the paper more appropriate for a general audience, we have added some
brief sentences giving some background in the introduction section.

2. The arxiv version of ref 17 by Cassar et al., which was included in our original
manuscript has now been published as Journal of Non-Crystalline Solids 547
(2020) 120297. Since the conclusions and details are different from those in the
original arxiv version, we have updated our discussion of this reference in the
introduction section of our manuscript.

3. Since we present studies only of the low temperature nucleation anomaly in the
$5\text{BaO}\cdot 8\text{SiO}_2$ glass we have changed the title, abstract and conclusion to reflect
this. As emphasized, however, we feel that our results will serve as a catalyst for
other checks on the same low-temperature anomalies previously reported in other
silicate glasses.

4. In our Supplementary Information file, we have added a section discussing how to
calculate the diffusion coefficient from the induction time.

Turning now to each referee in turn, we write the referee comments/questions in
italics and our responses in normal font.

SECOND ROUND: these changes improved the manuscript

Response to Reviewer #1:

1. Summary of the key results: Extensive crystal number density data were obtained at
a temperature 50K below that of maximum nucleation. The experimental
nucleation rate combined with values at different temperatures (from a previous
work), indicated that a theoretical model (CNT) describes the nucleation kinetics in
the whole T range, without a break that has been suggested by other researchers,
and also by these same authors in previous works.

Thank you for this comment. We agree that the possibility "without a break" has been
suggested by other researchers (ref. 17), but this has not been proven in any glass
studied. Our results presented are the first to definitively show that the break is an
artifact in the glass that we have studied. The reference to the arxiv article of the
previous study (ref. 17 in our initial manuscript) has now been updated by the authors
on the arxiv and published in the Journal of Non-Crystalline Solids (547 (2020)
120297). The conclusion in their published paper is different than that in the initial

arxiv version. The authors conclude that the statistical analysis that they applied to the
existing data in the literature could not establish whether the low-temperature behavior
was real or an artifact. In the discussion section they state "In the end, having
discussed the procedures and the current results in this communication, we are not able
to draw a definitive conclusion regarding the existence or not of the nucleation "break."

Our results, however, cast reasonable doubt about whether the "break" is a
phenomenon that happens for all oxide glasses. A recent report by Xia et al. [59], with
new measurements of the nuclei density in a Ba5Si8O21 glass at 50 K below T_{max},
supports our conclusion that the nucleation "break" is an artifact." The conclusion of
their JNCS ends with the statement, "With this setup, our results cast reasonable doubt
on whether the nucleation "break" is a universal phenomenon that happens for all oxide
glasses." Their study only casts doubt on whether the low-temperature anomaly was an
artifact or not. Our work that was submitted to Nature Materials provides the proof.
It is the first study to definitively show that the anomaly in the steady-state nucleation
rate is in fact an artifact in the glass we studied, and raises serious questions of whether
it is an artifact in other glasses that show the same low-temperature anomaly.

SECOND ROUND: After reading the latest version of the Cassar paper
(<https://arxiv.org/abs/1902.03193v2> , July, 2020), it became clear to me that it demonstrates
that the break does not exist in 4 out of 6 analyzed systems (Figs. 6
and 7). Moreover, for the other 2 glasses, they found only weak evidence for a
possible break of CNT. Therefore, Cassar concluded that ..., "our results
cast reasonable doubt on whether the nucleation "break" is a universal
phenomenon that happens for all oxide glasses." In other words, they could not generalize their
findings because there was no break in 4 glasses, but they were uncertain about two other
systems.

Hence, the Cassar paper already demonstrated that the break does not exist
in 4 systems, whereas in the current manuscript, Xia et al. reached the same
conclusion using a fifth system. Anyway, this is a good result.

To avoid confusion, I am copying Cassar's abstract below from arXiv:

<https://arxiv.org/abs/1902.03193v2>

[Submitted on 8 Feb 2019 (v1), last revised 20 Jul 2020 (this version, v2)]

Critical assessment of the alleged failure of the Classical Nucleation Theory at low temperatures -
Daniel Roberto Cassar et al

"The Classical Nucleation Theory allegedly fails to describe the temperature dependence of the
homogeneous crystal nucleation rates below the temperature of maximum nucleation, T_{max}.
Possible explanations for this suspected breakdown have been advanced in the literature.
However, the simplest hypothesis has never been tested, that it is a byproduct of nucleation
datasets that have not reached the steady-state regime. In this work, we tested this possibility by
analyzing published nucleation data for oxide supercooled liquids, using only nucleation and
viscosity data measured in samples of the same glass batch that also have satisfied a steady-state
regime test. Furthermore, all the uncertainty and regression confidence bands were computed and
considered. Having this rigorous protocol, among the 6 datasets analyzed, we only found weak
evidence supporting the existence of the nucleation break in 2 datasets. Our collective results thus
indicate that the break at T_{max} is not a common feature of all glass-formers."

Their Figures 6 and 7 show not break for several of the analyzed systems.

2. Originality and significance:

I have read some of the cited articles, including an ARXIV document from the San
Carlos group. I have also searched the Web-of-Science and Scopus for recent
journal publications on this problem, but did not find any. So, I don't know whether

the San Carlos group has finally published that article. The title of the arxiv doc by
Casar et al. is "The failure of the Classical Nucleation Theory at low temperatures
resolved" <https://arxiv.org/ftp/arxiv/papers/1902/1902.03193.pdf> In that arxiv
paper, the authors selected and analysed nucleation rates for 6 glasses in
temperature ranges where they (claimed) have reached the steady-state regime.
Their conclusion was: "With this strategy, we proved that the alleged nucleation
break is indeed an experimental artifact! This result ends a four decade-old
dilemma and corroborates the use of CNT for analyses of crystal nucleation rates."
And, indeed, their figures 5a-f and 6a-f seem to support their conclusion. So, the
conclusion of this manuscript is exactly the same of the arxiv paper?
(<https://arxiv.org/ftp/arxiv/papers/1902/1902.03193.pdf>) The only difference is
that another glass was used. But Casar already demonstrated this same fact for 6
glasses.

As we mentioned in our reply to the first comment from the reviewer, the initial arxiv of
Cassar et al. from the San Carlos group has been updated in the arxiv and has now been
published as (Journal of Non-Crystalline Solids 547 (2020) 120297). A close
examination of the published paper (not the initial arxiv version) shows a different
conclusion than in the initial arxiv version. The authors conclude that the statistical
analysis that they applied to the existing data in the literature could not establish
whether the low-temperature behavior was real or an artifact. Our study is the first one
to definitively show that it is an artifact for the glass we studied. Since the same
anomaly has been reported in other silicate glasses, our work indicates that these are
also likely artifacts and should be reinvestigated.

In our revised manuscript, we have updated the reference to the Cassar paper from
that of the initial arxiv to the published paper (Journal of Non-Crystalline Solids 547
(2020) 120297), which contains a different discussion and conclusion from the initial

arxiv version. We have also updated our manuscript to their work: "Recently, without
being able to draw a definitive conclusion, Cassar et al.²⁰ also cast reasonable doubt
about whether the low-temperature anomaly is a real phenomenon. Their
conclusion was based on a statistical analysis of existing published data on several
silicate glasses, but contained no firm demonstration in any glass. No new direct
measurements were made to demonstrate that the steady-state nucleation had indeed
been obtained below the peak nucleation temperature."

SECOND ROUND: Based on published extensive experimental data from different authors, above
and below T_g , Cassar firmly demonstrated that the CNT break is not seen in 4 systems (out of 6
analysed). The current manuscript of Xia et al. confirms those findings with another glass.

3. Data & methodology: validity of approach, quality of data, quality of presentation-
A standard approach was used: two-step crystallization, followed by optical
analyses of the microstructures. Then, they used the Kaischiev expression to
analyse the number density vs. time curves. This is also a well-known, accepted
procedure in this field.

Since there is no question raised by the reviewer in this comment, there is no change in
the revised manuscript on this point. The aim of our work was not to develop a new
methodology, but to use a tested method to examine the anomalous behavior that has
been reported in silicate glasses at low temperature.

SECOND ROUND: OK

4. The use of the Turnbull equation for the $D_g(T)$ calculation is an approximation,

which is only valid only under certain conditions, for some substances. I am not
sure whether it is valid for this bariumsilicate material. Perhaps the authors should
also test their results with other approximations for D_g .

The reviewer is correct; the Turnbull approximation assumes that the heat capacities of
the liquid and glass are the same. If the heat capacity difference between glass and
crystal as a function of temperature were known, a more accurate thermodynamic
driving free energy could be obtained. Unfortunately, these heat capacities are not
known. However, even if they were known, the magnitude of the thermodynamic
driving free energy would still increase approximately linearly with decreasing
temperature over the range of nucleation temperatures studied and would not change
the direction of the trend in the interfacial free energy or the work of critical cluster
formation. So, the key conclusion in our manuscript would remain the same.

SECOND ROUND: OK, thank you

5. They show error bars in all figures and Figure 4 shows the 95% confidence limits.
However, I should stress that the smooth curve through the red points (previous
data above the maximum) and black point (new data point at 50K below the
maximum) do not match.

In the text describing Figure 4 we state, "The data point at 1048 K falls outside of the
higher limit of the confidence bounds. This likely is an artifact of the fit, however.
The induction time was not measurable at this temperature; instead, it was estimated
from the data at 998 K, 1011 K and 1023 K." Further, the solid curve doesn't go exactly

through the data points because we used the linear fit to the interfacial free energy to
obtain a continuous curve, which will not go through the data exactly.

SECOND ROUND: OK

6. They could also include the confidence limits of the fit in Fig.3 (not mandatory)

We do not feel that it is necessary to add confidence bands for any type of fitting in Fig.
3 because it will not provide any additional useful information for the final conclusion.
Confidence bands are a statistical tool used to examine the fit for an assumed fitting
function. By fitting to different fitting functions, the confidence bands can be used as
a tool to allow a determination of the best function. They do not provide information
about the original data. Since in each subplot of Fig. 3 there is only one assumed
function, no information can be gained about the best function that matches the data.
With or without confidence bands the final conclusions are clear and the same.

SECOND ROUND: OK if one accepts that a single datapoint, without a rigorous statistical analysis
is sufficient to reach such important conclusion. Although, given the results of Cassar for 4 other
systems, this indeed seems to be the case. This reviewer is convinced that this break does not
exist, as demonstrated by Cassar et al and Xia et al

7. it resulted from only ONE new data point (at 50K below the maximum), so at least a
test in another low T should be relevant.

While it would be useful to obtain other low temperature measurements, due to the
COVID-19 pandemic this was not possible. These are extremely long experiments,
with each new measurement at a given low temperature nucleation temperature

requiring many months. The new nucleation measurement at the lowest temperature
reported in this glass clearly demonstrates that the low temperature anomaly previously
reported by us for this glass is incorrect. It is also likely incorrect in the other silicate
glasses where the anomaly has been reported.

SECOND ROUND: This reviewer is fully aware of the COVID-19 problem. However, another data point at a somewhat higher temperature, which would not take that long to collect, would be essential to confirm their conclusions, which is based on a single nucleation rate datapoint.

8. this study with one particular glass corroborates the results of a much stronger
article (arxiv 2019) that analysed 6 glasses, and has reached exactly the same
conclusion.

We have discussed this in an earlier response. The initial conclusion proposed by us
Cassar et al. from the San Carlos group in their original arxiv paper has changed in the
current arxiv entry and in their published Journal of Non-Crystalline Solids article. In
those later articles, they conclude that they were not able to reach a definitive
conclusion regarding the low-temperature anomaly in the nucleation rate. So, their
article was not stronger as stated by the referee. Their analysis suggested that the
anomaly was an artifact. Our data are the first to definitively demonstrate that it is an
artifact for the glass we studied, and raises the serious question of whether it is an
artifact in other glasses that show the same low-temperature anomaly. Since we have
not yet been able to confirm this in other glasses due to the COVID-19 pandemic,
impacting our laboratory studies and our colleagues at Corning Glass Inc., who were
going to furnish the samples but were abruptly required to work remotely before the
glasses could be prepared and characterized, we have changed the title, abstract and
conclusion in our submitted paper to focus on the $5\text{BaO}\cdot 8\text{SiO}_2$ glass. Our future
studies will investigate whether it is also an anomaly in other silicate glasses.

SECOND ROUND: Based on published experimental data from different authors, Cassar demonstrated that the break does not appear in 4 systems (out of 6 analysed), Figures 6 and 7. Hence, the present results confirm Cassar's findings.

9. Finally, I do not understand the several black squares in Figs. 3 and 4 labelled "this
work". I mean, only the point at the lowest T was obtained and should be referred as
to "this work".

In this work, although only one point reflects the new measurements of the number of
nuclei as a function of time at the nucleation temperature, all four black data points in
Figs. 3 and 4 were recalculated and updated in this work's analysis section (details
can be seen in the supplementary methods section in the Supplementary Information
file). So, in this work, for the outcome after measurement and analysis, there are four
points instead of one point.

SECOND ROUND: OK

10. I believe the use of a linear fitting using only the last data points influence both the
value of the resulting steady-state nucleation rate and the θ ? I believe the fitting
procedure should be made with the Kashchiev model using ALL points of the $N_v(t)$
curve. Apparently, the red points in the beginning were completely neglected?

The linear fitting method we used is a widely accepted method to obtain the
steady-state nucleation rates and the induction times. That there are other ways to
obtain the result doesn't invalidate this approach. Further, while the Kashchiev
expression is indeed the best analytical expression (see "Transient Nucleation in
Condensed Systems," K. F. Kelton, A. L. Greer and C. V. Thompson, Journal of
Chemical Physics, 79, 6261 (1983)), it is still based on assumptions. We feel that the
standard method of analyzing the nucleation data is the best approach.

SECOND ROUND: The problem in using only the final data points – which seem to be in a straight line - rather than the whole N vs. time curve, is that you are implicitly assuming that steady-state conditions have already been reached.

11. Also, Schneidman has shown long ago that there is an interdependence between the
true nucleation induction times, θ , and $t_{(ind_d)}$, hence a time shift, t_s , should
be considered in the model for properly fitting these curves. And these true θ s
should be used in the CNT plots. Please comment.

There are several ways to obtain the correct induction time (at n^* for the nucleation
temperature) from the measured data. We have chosen to use the Shneidman-Weinberg
expression, the validity of which has been confirmed (Nucleation in Condensed Matter
– Applications in Materials and Biology, K. F. Kelton and A. L. Greer, Pergamon
Materials Series, Elsevier, Amsterdam (2010)). That is why we corrected the
measured induction times, which were for n^* at the growth temperature, to obtain the
induction time for n^* for the nucleation temperature. As discussed on page 81, figure 11
of chapter 3 in "Nucleation in Condensed Matter – Applications in Materials and
Biology," K. F. Kelton and A. L. Greer, Elsevier (2010), this approach gives a valid
correction to the data so that the standard Kashchiev treatment can then be used. Both
the induction time correction and the standard Kashchiev treatment have been
embedded in equation S-1 in the Supplementary Information file. Details about the
derivations of equation S-1 can be seen in Journal of Non-Crystalline Solids 525 (2019)
119575.

SECOND ROUND: Ok, let's leave it this way. However, please keep in mind that nucleation in a glass (below T_g) is further complicated due to the glass relaxation's interference on the nucleation pathways and kinetics. Nucleation above T_g takes place in a SCL, whereas below T_g it takes place in a glass. This is likely the reason why you cannot describe your N vs. time curve with the Kash equation

12. The test of the proposed hypothesis (no break in the theory) was done using only
one temperature below the maximum (50 K). It would be good practice to double or
triple check this result by using at least another low enough temperature. In other
words, this study would be much stronger if it were extended to, at least, another
temperature (say, 973 K or 985 K), for times long enough to reach the steady-state.

Unfortunately, due to the COVID-19 pandemic, we are not in a position to supply
another low T measurement, which is a very long experiment. But, we maintain that the
data we have obtained in the manuscript clearly demonstrate that in this glass the low
temperature anomaly in the steady-state nucleation rate previously reported is
incorrect.

SECOND ROUND: OK, we have already discussed this issue

13. The points in Figure 3 (d) are not well aligned; it would be educational to include
the confidence interval of the adjustments, as you did in Figure 4.

The linear fit doesn't have to go through the data points exactly. Currently, Figure 3d is
good and clear in presenting the trend observed in this work. So, we don't think it is
necessary to add confidence bands for the linear fit. The detailed reasons are the same
as our response to the 6th comment. (One minor note: the reason we do include
confidence bands in Fig. 4 is as a guide to the eye; it can better differentiate the trends
of the data in this work from that of the previous work. But in Fig. 3d, confidence bands
are not necessary for this differentiation.

SECOND ROUND: OK

14. The use of the Kashchiev equation (Reference 27) for the calculation of the
diffusion coefficient from the induction times is commented, but the equation is not
shown.

To address this comment, we have added the Kashchiev equation into the
Supplementary Information file. See the section named as "Using the Kashchiev
expression to calculate the diffusion coefficient from the induction time for the critical
size at the nucleation temperature".

SECOND ROUND: OK, thank you

15. The development temperature, T_d (1073 K), used for the some data points of the
lowest temperature analyzed is different from the T_d (1119 K) used for the first
data points and also for all the other temperatures. The time shift, t_s , is a function of
T_d , which also affects the $t_{ind,d}$ that was determined from the linear
adjustment of the last points and, consequently, the $t_{ind,n}$ determined by
equation S-3. The authors should comment on how the use of these different T_d
would affect their calculations?

The Kashchiev treatment can only be used for the induction time for the critical
cluster size at the nucleation temperature. That is why we corrected the measured

induction times (which were for the critical cluster size at the growth temperature)
using the expression developed by Shneidman and Weinberg. As discussed on page 81,
figure 11 of chapter 3 in "Nucleation in Condensed Matter – Applications in Materials
and Biology," K. F. Kelton and A. L. Greer, Elsevier (2010), this approach gives a
valid correction to the data so that the standard Kashchiev treatment can then be used.
Both the correction and the standard Kashchiev treatment have been embedded in
equation S-1 in the Supplementary Information file. Details of the derivations of
equation S-1 can be found in Journal of Non-Crystalline Solids 525 (2019) 119575.

The different growth temperatures have already been included in our calculations.
Specifically, when we input the nucleation data at 948 K into equation S-1, we also
input S-1 the related growth temperature, 1073 K. When we input the nucleation data
at 998 K, 1011 K, and 1023 K into equation S-1, the related growth temperature, 1119
K is also input. The details about the iterative calculation steps can be found in the
analysis steps part of the Supplementary Methods in the Supplementary Information
file.

SECOND ROUND: OK, thank you.

Minor comments

The title is considerably long but more appropriate than the previous one since the test was performed only on one glass. Data for this particular Ba-silicate glass are available just in publications of the Kelton group.

Introduction: "A barium-silicate glass was chosen since they have higher nucleation rates than other glasses, such as lithium disilicate or soda-lime silicate so that it takes less time to obtain a significant number of nuclei.": Some soda-lime silicate glasses may have higher nucleation rates (e.g., the 2-1-3 glass) than barium silicates.

Results, line 5: It would need to clarify that it refers to the induction time at the growth temperature (as written in line 7, page 6), since this is the first time this variable is mentioned, and this is not a specific glass science journal.

Conclusion: "For the new measurements reported here the glasses were given a much longer nucleation treatment than was used in all previous measurements. These new data do not show a low-temperature anomaly": This paragraph is somewhat confusing; it implies that various nucleation rates at different temperatures were determined.

The used heating rates of the double stage heat treatment and their possible effects on IST were not reported.

Measurement of possible athermic nuclei were not reported.

Overall this is a very good paper, with a few problems.

Reviewer #3 (Remarks to the Author):

Glasses have accompanied and facilitated human civilization in the last 2000 years and are indispensable to modern communication and computation apparatus. The amazing optical, thermodynamic, mechanical, and electronic properties of glasses are extensively researched, however, the prime issue of the origin of glasses, i.e., why some melts fail to crystallize upon cooling, is still elusive. A classical explanation from the 1950s relied on crystal nucleation theory and posited that at lower temperatures the viscosity of melts of complex molecules increased so much that it suppressed crystal nucleation. An important open question was whether the lower temperatures also enforced a higher nucleation barrier. Numerous determinations of nucleation rates in glass forming melts, aptly summarized in this manuscript, appear to furnish a positive answer to this question. This outcome implies that classical nucleation theory does not apply to glass forming melts and elaborate non-classical phenomena dominate the kinetic pathways. Xia et al. design a technique which allows them to maintain stable temperature for months on end and in an experimental tour de force monitor the time evolution of the crystal nucleation rate in a judiciously chosen glass. This approach convincingly demonstrates that the investigations that demonstrate increasing nucleation barriers at lower temperatures have ignored a second corollary of the high viscosity, the exceedingly slow egress of the nucleation process to a steady state, manifested as a nucleation delay time longer than expected by orders of magnitude. In consequence, the monitored crystal nucleation events were not in a steady regime and, accordingly, their rates were substantially slower than the predictions of classical theory. The slow measured nucleation rates were misinterpreted as an outcome of higher nucleation barriers. Xia et al. clearly demonstrate that classical nucleation theory applies to glass forming melts and potential alternative nucleation pathways are not in play.

The study is well conceived and perfectly executed, the paper is carefully written, the results are placed in the context of classical theory and the basic assumptions that appeared violated by

previously collected data are properly stated. Particularly appealing is Fig. 3, which displays three tests of the compliance of the measured nucleation behaviors to CNT.

Owing to its huge fundamental and applied merit and the quality of the investigation, I certainly think that the manuscript by Xia et al. should be published in Nature Communications. I suggest a few mostly editorial revisions that will significantly enhance the impact of this paper.

In the Introductions, please specify that previously measured nucleation rates below a certain temperature were significantly lower than the prediction of CNT based on assuming slower kinetics due to greater viscosity at the lower temperatures. This discrepancy enforced the assumption that the nucleation barrier shoots up. This explanation will tie the misinterpretation of the previous results with your powerful demonstration that the nucleation rates measured before steady state is reached are lower.

Along the same line of thought, please use Ref. 28 or the more recent book by Kashchiev to explain why nucleation rates measured before steady state is reached are slower and not faster than the steady state values; this transition imposes a superlinear correlation $N(t)$, so nicely demonstrated in Fig. 2.

p. 4: please denote the increase of the number of crystals that nucleate per unit volume per unit time as superlinear instead of the more generic nonlinear. Do not use parentheses to bracket the values of I_{st} and θ . Please report the values of I_{st} and θ with fewer significant digits as demanded by the standard deviations of the respective measurements. Consult a book on statistics of experiment if needed.

Please use same units of volume and time throughout. Volume is now reported in cubic microns and cubic millimeters, whereas time is measured seconds, minutes and even days. This makes verifying the stated parameter values with the presented data confusing.

In Fig. 3b, the legends that state growth T 1093K and growth T 1119K in both black and red are confusing. It is not clear if the points belong to the figure legend or to the linear correlation starting at lower temperatures.

Peter G. Vekilov, University of Houston

We thank the referees for a second round review of our manuscript. To better acknowledge that the analysis by Cassar et. al. motivated us to carry out the experiment outlined in our manuscript, we have changed our discussion in the introduction section. We received second round reviews only from Reviewers #1 and #3; our responses follow. The paragraph starting with *SECOND ROUND COMMENT* in italics font are the reviewers' comments in the second round, and the paragraph beginning with *SECOND ROUND RESPONSE* in normal font are our related responses. For completeness, we also include the first round comments and our first round responses when they are related to second round comments.

Response to Reviewer #1:

1. *FIRST ROUND COMMENT: Summary of the key results: Extensive crystal number density data were obtained at a temperature, 50K below that of maximum nucleation. The experimental nucleation rate combined with values at different temperatures (from a previous work), indicated that a theoretical model (CNT) describes the nucleation kinetics in the whole T range, without a break that has been suggested by other researchers, and also by these same authors in previous works.*

FIRST ROUND RESPONSE: Thank you for this comment. We agree that the possibility “without a break” has been suggested by other researchers (ref. 17), but this has not been proven in any glass studied. Our results presented are the first to definitively show that the break is an artifact in the glass that we have studied. The reference to the arxiv article of the previous study (ref. 17 in our initial manuscript) has now been updated by the authors on the arxiv and published in the *Journal of Non-Crystalline Solids* (547 (2020) 120297). The conclusion in their published paper is different than that in the initial arxiv version. The authors conclude that the statistical analysis that they applied to the existing data in the literature could not establish whether the low-temperature behavior was real or an artifact. In the discussion section they state “In the end, having discussed the procedures and the current results in this communication, we are not able to draw a definitive conclusion regarding the existence or not of the nucleation “break.” Our results, however, cast reasonable doubt about whether the “break” is a phenomenon that happens for all oxide glasses. A recent report by Xia et al. [59], with new measurements of the nuclei density in a Ba₅Si₈O₂₁ glass at 50 K below T_{max}, supports our conclusion that the nucleation “break” is an artifact. The conclusion of their JNCS ends with the statement, “With this setup, our results cast reasonable doubt on whether the nucleation “break” is a universal phenomenon that happens for all oxide glasses.” Their study only casts doubt on whether the low-temperature anomaly was an artifact or not. Our work that was submitted to *Nature Materials* provides the proof (*note*, this review was for our original submission of this article to *Nature Materials*; it was transferred to *Nature Communications*). It is the first study to definitively show that the anomaly in the steady-state nucleation rate is

in fact an artifact in the glass we studied, and raises serious questions of whether it is an artifact in other glasses that show the same low-temperature anomaly.

SECOND ROUND COMMENT: *After reading the latest version of the Cassar paper (<https://arxiv.org/abs/1902.03193v2> , July, 2020), it became clear to me that it demonstrates that the break does not exist in 4 out of 6 analyzed systems (Figs. 6 and 7). Moreover, for the other 2 glasses, they found only weak evidence for a possible break of CNT. Therefore, Cassar concluded that ..., "our results cast reasonable doubt on whether the nucleation "break" is a universal phenomenon that happens for all oxide glasses." In other words, they could not generalize their findings because there was no break in 4 glasses, but they were uncertain about two other systems.*

Hence, the Cassar paper already demonstrated that the break does not exist in 4 systems, whereas in the current manuscript, Xia et al. reached the same conclusion using a fifth system. Anyway, this is a good result.

To avoid confusion, I am copying Cassar's abstract below from arXiv: <https://arxiv.org/abs/1902.03193v2> [Submitted on 8 Feb 2019 (v1), last revised 20 Jul 2020 (this version, v2)] Critical assessment of the alleged failure of the Classical Nucleation Theory at low temperatures - Daniel Roberto Cassar et al "The Classical Nucleation Theory allegedly fails to describe the temperature dependence of the homogeneous crystal nucleation rates below the temperature of maximum nucleation, T_{max} . Possible explanations for this suspected breakdown have been advanced in the literature. However, the simplest hypothesis has never been tested, that it is a byproduct of nucleation datasets that have not reached the steady-state regime. In this work, we tested this possibility by analyzing published nucleation data for oxide supercooled liquids, using only nucleation and viscosity data measured in samples of the same glass batch that also have satisfied a steady-state regime test. Furthermore, all the uncertainty and regression confidence bands were computed and considered. Having this rigorous protocol, among the 6 datasets analyzed, we only found weak evidence supporting the existence of the nucleation break in 2 datasets. Our collective results thus indicate that the break at T_{max} is not a common feature of all glass-formers."

Their Figures 6 and 7 show not break for several of the analyzed systems.

SECOND ROUND RESPONSE: We feel that this is a misinterpretation of the Cassar publication *Journal of Non-Crystalline Solids* **547** (2020) 120297. The fact is that in Cassar et al. there are 6 datasets, not 6 compositions; 3 datasets are for the same composition, $\text{Li}_2\text{Si}_2\text{O}_5$. In Cassar et. al, the key results are shown in Figs. 6 and 7 of their paper. In Fig. 6a, for the first $\text{Li}_2\text{Si}_2\text{O}_5$ dataset, 5 out of the 7 data points below the peak nucleation temperature T_{max} were not considered in their analysis because they did not pass their data quality test standards. And the 2 data points at low temperature that did pass their standards showed no nucleation anomaly below T_{max}

for this glass. In Fig. 6b, for the second Li₂Si₂O₅ dataset, one low-temperature point below T_{max} failed to meet their standards and two low-temperature points below T_{max} showed no nucleation anomaly. In Fig 6c, for the third Li₂Si₂O₅ dataset, there is one data point below T_{max}; the value of that point is slightly below or approximately equal to the T_{max} point in Fig. 6c and its interfacial free energy is larger than the value at T_{max}, as shown in Fig. 7c. Figures 6c and 7c suggest that there is a nucleation anomaly below T_{max} in Li₂Si₂O₅. However, it might be argued for Fig. 6c that the low temperature point is still within the confidence band for linear fitting. It should be emphasized that the 95% confidence band for linear regression is a statistical evaluation tool, used when a linear relationship is assumed. The band will differ for other functional forms (i.e. nonlinear). If the change of trend from the T_{max} point to the low-temperature point in Fig. 6c is correct an assumed linear relationship would be incorrect for calculating the confidence band.

Also, in their paper Cassar et al. themselves point to the third dataset for the composition Li₂Si₂O₅ as the supporting evidence existence of nucleation break. They state - “Subplot 6c, however, is the first result for which we cannot completely reject the presence of the nucleation “break”.” In another paragraph they state - “In the subplots of Fig. 7, we seek evidence for the nucleation “break” in the form of a change in the monotonic growth of σ regarding the temperature. If we consider only the data points that passed the steady-state test, we observe that only subplot 7c may have a non-monotonic growth of σ .”

It might be asked why of the three datasets of Li₂Si₂O₅ analyzed via the approach suggested by Cassar et al. two datasets showed no low-temperature anomaly and one dataset only suggested a low-temperature nucleation anomaly? We agree that the approach used by Cassar et al. does suggest that the break might not exist, but cannot confirm the non-existence of the break. Based on this weakness can it be unequivocally demonstrated or conclusively confirmed that the nucleation anomaly does not exist in any composition analyzed by Cassar et. al? We feel that based on their Figs. 6 and 7 and their own stated conclusions, the answer is no.

Furthermore, both you and we agree that Cassar et al. is based on an analysis of previously published experimental data from different authors. In fact, this is not the first time researchers have used previously published data to try to find an answer about the low temperature nucleation anomaly. In *J. Chem. Phys.* 145, 211920 (2016), Gupta et al. also analyzed the Li₂Si₂O₅ glass, using previous data in the literature. Shown in Figs. 2 and 3 of their paper, their analysis of a more complete dataset of Li₂Si₂O₅ from the literature indicates the existence of a low-temperature nucleation anomaly in this glass, i.e. reaching a different conclusion from that of Cassar et al. Further, in *J. of Non-Cryst. Solids*, 447 (2016) 35–44, Fokin et al. analyzed previously taken literature data for Na₄CaSi₃O₉ and Na₂Ca₂Si₃O₉ glasses. As argued from Fig. 1 of their paper their analysis indicates a low-temperature nucleation anomaly in both glasses, again a finding that is different from Cassar et al’s study of Na₄CaSi₃O₉ and Na₂Ca₂Si₃O₉.

In summary, in the analysis of previously published nucleation data, different researchers have reached opposite conclusions. This is the reason that although we certainly respect the work by Cassar et al., which is based on a more strict assessment of the data quality, it does not present a firm conclusion, nor a proof regarding the nucleation anomaly question for any of the glasses that they have studied.

We gratefully acknowledge, however, that the analysis by Cassar et. al. partially motivated us to carry out the experiment outlined in our manuscript. We have modified the discussion in the introduction to emphasize as follows.

“Already in some previous studies (Zanotto et al.¹⁸ and Greer et al.¹⁹) the possibility was raised that the nucleation anomaly might be an artifact, but without providing conclusive evidence. A series of previously published nucleation data sets were recently re-analyzed by Cassar et al.²⁰, focusing on data near the peak nucleation temperature. They concluded that not all data points could be taken with equal confidence, finding variations even across data sets for the same type of glass. From this, they cast doubt on the widely studied nucleation anomaly. Partially motivated by the conclusions of Cassar et al.²⁰ and by those from other data analyses (such as Gupta et al.¹⁷) we concluded that the anomaly might be an artifact resulting from insufficient heating time at the low nucleation temperatures.

Here we show that the anomaly previously reported in a 5BaO·8SiO₂ glass¹⁰ was indeed an experimental artifact. This was demonstrated by using a suitably designed experimental procedure and tracking the nucleation process over extensively long periods of time ...”

2. *FIRST ROUND COMMENT: Originality and significance:*

I have read some of the cited articles, including an ARXIV document from the San Carlos group. I have also searched the Web-of-Science and Scopus for recent journal publications on this problem, but did not find any. So, I don't know whether the San Carlos group has finally published that article. The title of the arxiv doc by Casar et all. Is "The failure of the Classical Nucleation Theory at low temperatures resolved" <https://arxiv.org/ftp/arxiv/papers/1902/1902.03193.pdf> In that arxiv paper, the authors selected and analysed nucleation rates for 6 glasses in temperature ranges where they (claimed) have reached the steady-state regime. Their conclusion was: "With this strategy, we proved that the alleged nucleation break is indeed an experimental artifact! This result ends a four decade-old dilemma and corroborates the use of CNT for analyses of crystal nucleation rates." And, indeed, their figures 5a-f and 6a-f seem to support their conclusion. So, the conclusion of this manuscript is exactly the same of the arxiv paper? (<https://arxiv.org/ftp/arxiv/papers/1902/1902.03193.pdf>) The only difference is that another glass was used. But Casar already demonstrated this same fact for 6 glasses.

FIRST ROUND RESPONSE: As we mentioned in our reply to the first comment from the reviewer, the initial arxiv of Cassar *et al.* from the San Carlos group has been updated in the arxiv and has now been published as (*Journal of Non-Crystalline Solids* 547 (2020) 120297). A close examination of the published paper (not the initial arxiv version) shows a different conclusion than in the initial arxiv version. The authors conclude that the statistical analysis that they applied to the existing data in the literature could not establish whether the low-temperature behavior was real or an artifact. Our study is the first one to definitively show that it is an artifact for the glass we studied. Since the same anomaly has been reported in other silicate glasses, our work indicates that these are also likely artifacts and should be reinvestigated.

In our revised manuscript, we have updated the reference to the Cassar paper from that of the initial arxiv to the published paper (*Journal of Non-Crystalline Solids* **547** (2020) 120297), which contains a different discussion and conclusion from the initial arxiv version. We have also updated our manuscript to their work: “Recently, without being able to draw a definitive conclusion, Cassar *et al.*²⁰ also cast reasonable doubt about whether the low-temperature anomaly is a real phenomenon. Their conclusion was based on a statistical analysis of existing published data on several silicate glasses, but contained no firm demonstration in any glass. No new direct measurements were made to demonstrate that the steady-state nucleation had indeed been obtained below the peak nucleation temperature.”

SECOND ROUND COMMENT: *Based on published extensive experimental data from different authors, above and below T_g, Cassar firmly demonstrated that the CNT break is not seen in 4 systems (out of 6 analysed). The current manuscript of Xia et al. confirms those findings with another glass.*

SECOND ROUND RESPONSE: We have already addressed this comment in our Second Round response to the comment No. 1.

3. **FIRST ROUND COMMENT:** *Data & methodology: validity of approach, quality of data, quality of presentation- A standard approach was used: two-step crystallization, followed by optical analyses of the microstructures. Then, they used the Kaischiev expression to analyse the number density vs. time curves. This is also a well-known, accepted procedure in this field.*

FIRST ROUND RESPONSE: Since there is no question raised by the reviewer in this comment, there is no change in the revised manuscript on this point. The aim of our work was not to develop a new methodology, but to use a tested method to examine the anomalous behavior that has been reported in silicate glasses at low temperature.

SECOND ROUND COMMENT: *OK*

SECOND ROUND RESPONSE: Thanks

4. *FIRST ROUND COMMENT: The use of the Turnbull equation for the $D_g(T)$ calculation is an approximation, which is only valid only under certain conditions, for some substances. I am not sure whether it is valid for this barium silicate material. Perhaps the authors should also test their results with other approximations for D_g .*

FIRST ROUND RESPONSE: The reviewer is correct; the Turnbull approximation assumes that the heat capacities of the liquid and glass are the same. If the heat capacity difference between glass and crystal as a function of temperature were known, a more accurate thermodynamic driving free energy could be obtained. Unfortunately, these heat capacities are not known. However, even if they were known, the magnitude of the thermodynamic driving free energy would still increase approximately linearly with decreasing temperature over the range of nucleation temperatures studied and would not change the direction of the trend in the interfacial free energy or the work of critical cluster formation. So, the key conclusion in our manuscript would remain the same.

SECOND ROUND COMMENT: *OK, thank you*

SECOND ROUND RESPONSE: Thanks

5. *FIRST ROUND COMMENT: They show error bars in all figures and Figure 4 shows the 95% confidence limits. However, I should stress that the smooth curve through the red points (previous data above the maximum) and black point (new data point at 50K below the maximum) does not match.*

FIRST ROUND RESPONSE: In the text describing Figure 4 we state, "The data point at 1048 K falls outside of the higher limit of the confidence bounds. This likely is an artifact of the fit, however. The induction time was not measurable at this temperature; instead, it was estimated from the data at 998 K, 1011 K and 1023 K." Further, the solid curve doesn't go exactly through the data points because we used the linear fit to the interfacial free energy to obtain a continuous curve, which will not go through the data exactly.

SECOND ROUND COMMENT: *OK*

SECOND ROUND RESPONSE: Thanks

6. *FIRST ROUND COMMENT: They could also include the confidence limits of the fit in Fig.3 (not mandatory)*

FIRST ROUND RESPONSE: We do not feel that it is necessary to add confidence bands for any type of fitting in Fig. 3 because it will not provide any additional useful information for the final conclusion. Confidence bands are a statistical tool used to examine the fit for an assumed fitting function. By fitting to different fitting functions, the confidence bands can be used as a tool to allow a determination of the best function. They do not provide information about the original data. Since in each subplot of Fig. 3 there is only one assumed function, no information can be gained about the best function that matches the data. With or without confidence bands the final conclusions are clear and the same.

SECOND ROUND COMMENT: *OK if one accepts that a single datapoint, without a rigorous statistical analysis is sufficient to reach such important conclusion. Although, given the results of Cassar for 4 other systems, this indeed seems to be the case. This reviewer is convinced that this break does not exist, as demonstrated by Cassar et al and Xia et al.*

SECOND ROUND RESPONSE: Thank you for these comments. Although an additional temperature point could be useful, one temperature point as one counter example is sufficient to prove that the previous general statement of low-temperature nucleation anomaly is wrong in this glass. Concerning the statistical analysis, we have already included the necessary error analysis and we have already replied to the confidence band comment in our First Round response. Regarding Cassar et al., we have addressed this comment in our Second Round response to the comment No. 1.

7. **FIRST ROUND COMMENT:** *it resulted from only ONE new data point (at 50K below the maximum), so at least a test in another low T should be relevant.*

FIRST ROUND RESPONSE: While it would be useful to obtain other low temperature measurements, due to the COVID-19 pandemic this was not possible. These are extremely long experiments, with each new measurement at a given low temperature nucleation temperature requiring many months. The new nucleation measurement at the lowest temperature reported in this glass clearly demonstrates that the low temperature anomaly previously reported by us for this glass is incorrect. It is also likely incorrect in the other silicate glasses where the anomaly has been reported.

SECOND ROUND COMMENT: *This reviewer is fully aware of the COVID-19 problem. However, another data point at a somewhat higher temperature, which would not take that long to collect, would be essential to confirm their conclusions, which is based on a single nucleation rate datapoint.*

SECOND ROUND RESPONSE: Thank you for this point. As we said in our First Round response, the long-time nucleation measurement at the lowest temperature reported in this glass clearly demonstrates that the low-temperature anomaly previously reported for this glass is incorrect.

8. *FIRST ROUND COMMENT: this study with one particular glass corroborates the results of a much stronger article (arxiv 2019) that analysed 6 glasses, and has reached exactly the same conclusion.*

FIRST ROUND RESPONSE: We have discussed this in an earlier response. The initial conclusion proposed by Cassar *et al.* from the San Carlos group in their original arxiv paper has changed in the current arxiv entry and in their published Journal of Non-Crystalline Solids article. In those later articles, they conclude that they were not able to reach a definitive conclusion regarding the low-temperature anomaly in the nucleation rate. So, their article was not stronger as stated by the referee. Their analysis suggested that the anomaly was an artifact. Our data are the first to definitively demonstrate that it is an artifact for the glass we studied, and raises the serious question of whether it is an artifact in other glasses that show the same low-temperature anomaly. Since we have not yet been able to confirm this in other glasses due to the COVID-19 pandemic, impacting our laboratory studies and our colleagues at Corning Glass Inc., who were going to furnish the samples but were abruptly required to work remotely before the glasses could be prepared and characterized, we have changed the title, abstract and conclusion in our submitted paper to focus on the $5\text{BaO}\cdot 8\text{SiO}_2$ glass. Our future studies will investigate whether it is also an anomaly in other silicate glasses.

SECOND ROUND COMMENT: Based on published experimental data from different authors, Cassar demonstrated that the break does not appear in 4 systems (out of 6 analysed), Figures 6 and 7. Hence, the present results confirm Cassar's findings.

SECOND ROUND RESPONSE: Thank you for this comment. We have already addressed this comment in our Second Round response to the comment No. 1.

9. *FIRST ROUND COMMENT: Finally, I do not understand the several black squares in Figs. 3 and 4 labelled "this work". I mean, only the point at the lowest T was obtained and should to referred as to "this work".*

FIRST ROUND RESPONSE: In this work, although only one point reflects the new measurements of the number of nuclei as a function of time at the nucleation temperature, all four black data points in Figs. 3 and 4 were recalculated and updated in this work's analysis section (details can be seen in the supplementary methods section in the Supplementary Information file). So, in this work, for the outcome after measurement and analysis, there are four points instead of one point.

SECOND ROUND COMMENT: OK

SECOND ROUND RESPONSE: Thanks

10. FIRST ROUND COMMENT: I believe the use of a linear fitting using only the last data points influence both the value of the resulting steady-state nucleation rate and the theta? I believe the fitting procedure should be made with the Kashchiev model using ALL points of the $N_v(t)$ curve. Apparently, the red points in the beginning were completely neglected?

FIRST ROUND RESPONSE: The linear fitting method we used is a widely accepted method to obtain the steady-state nucleation rates and the induction times. That there are other ways to obtain the result doesn't invalidate this approach. Further, while the Kashchiev expression is indeed the best analytical expression (see "Transient Nucleation in Condensed Systems," K. F. Kelton, A. L. Greer and C. V. Thompson, *Journal of Chemical Physics*, **79**, 6261 (1983)), it is still based on assumptions. We feel that the standard method of analyzing the nucleation data is the best approach.

SECOND ROUND COMMENT: The problem in using only the final data points – which seem to be in a straight line - rather than the whole N vs. time curve, is that you are implicitly assuming that steady-state conditions have already been reached.

SECOND ROUND RESPONSE: As we indicated in our First Round response, that there are other ways to obtain the result doesn't invalidate this widely accepted linear fitting approach.

11. FIRST ROUND COMMENT: Also, Schneidman has shown long ago that there is an interdependence between the true nucleation induction times, theta, and $t_{(ind_d)}$, hence a time shift, t_s , should be considered in the model for properly fitting these curves. And these true thetas should be used in the CNT plots. Please comment.

FIRST ROUND RESPONSE: There are several ways to obtain the correct induction time (at n^* for the nucleation temperature) from the measured data. We have chosen to use the Shneidman-Weinberg expression, the validity of which has been confirmed (*Nucleation in Condensed Matter – Applications in Materials and Biology*, K. F. Kelton and A. L. Greer, Pergamon Materials Series, Elsevier, Amsterdam (2010)). That is why we corrected the measured induction times, which were for n^* at the growth temperature, to obtain the induction time for n^* for the nucleation temperature. As discussed on page 81, figure 11 of chapter 3 in "*Nucleation in Condensed Matter – Applications in Materials and Biology*," K. F. Kelton and A. L. Greer, Elsevier (2010), this approach gives a valid correction to the data so that the standard Kashchiev treatment can then be used. Both the induction time correction and the standard Kashchiev treatment have been embedded in equation S-1 in the Supplementary Information file. Details about the derivations of equation S-1 can be seen in *Journal of Non-Crystalline Solids* 525 (2019) 119575.

SECOND ROUND COMMENT: *Ok, let's leave it this way. However, please keep in mind that nucleation in a glass (below T_g) is further complicated due to the glass relaxation's interference on the nucleation pathways and kinetics. Nucleation above T_g takes place in a SCL, whereas below T_g it takes place in a glass. This is likely the reason why you cannot describe your N vs. time curve with the Kash equation*

SECOND ROUND RESPONSE: This is a valid point. Structural relaxation below T_g could influence the kinetics and also, through the elastic strain energy, the thermodynamic work of cluster formation. Looking at the kinetics first; these are typically described in terms of the viscosity, which can be dramatically lower than the extrapolated value from the supercooled liquid and can change with structural relaxation. If the data were analyzed using the extrapolated value for the viscosity the results would be incorrect. The measured induction time, however, directly reflects the interfacial kinetics, which is the reason that we analyze our data using this. Structural relaxation could influence the induction time as well, but if that were the case over the measurement time we would not see a linear relation between the number of nuclei and time in the steady-state regime. This suggests that any change in the kinetics with structural relaxation occurs quickly at the nucleation temperature. This is consistent with numerical estimates that we have made based on the kinetic parameters. Regarding the work of cluster formation, the role of the elastic strain energy is insufficient to explain the anomalous break observed by others (see *Journal of Non-Crystalline Solids* 432 (2016) 325–333). Finally, we have never suggested that the Kashchiev expression is incorrect. Indeed, we have used an expression obtained from this treatment in our analysis. As discussed in our First Round Response, the Kashchiev expression was used to obtain an expression that we use to correct the induction time from the induction time for n^* at the growth temperature to the induction time for n^* at the nucleation temperature (see *Nucleation in Condensed Matter*, K. F. Kelton and A. L. Greer, Elsevier (2010), pages 76-77).

12. **FIRST ROUND COMMENT:** *The test of the proposed hypothesis (no break in the theory) was done using only one temperature below the maximum (50 K). It would be good practice to double or triple check this result by using at least another low enough temperature. In other words, this study would be much stronger if it were extended to, at least, another temperature (say, 973 K or 985 K), for times long enough to reach the steady-state.*

FIRST ROUND RESPONSE: Unfortunately, due to the COVID-19 pandemic, we are not in a position to supply another low T measurement, which is a very long experiment. But, we maintain that the data we have obtained in the manuscript clearly demonstrate that in this glass the low temperature anomaly in the steady-state nucleation rate previously reported is incorrect.

SECOND ROUND COMMENT: *OK, we have already discussed this issue*

SECOND ROUND RESPONSE: Thanks

13. FIRST ROUND COMMENT: The points in Figure 3 (d) are not well aligned; it would be educational to include the confidence interval of the adjustments, as you did in Figure 4.

FIRST ROUND RESPONSE: The linear fit doesn't have to go through the data points exactly. Currently, Figure 3d is good and clear in presenting the trend observed in this work. So, we don't think it is necessary to add confidence bands for the linear fit. The detailed reasons are the same as our response to the 6th comment. (One minor note: the reason we do include confidence bands in Fig. 4 is as a guide to the eye; it can better differentiate the trends of the data in this work from that of the previous work. But in Fig. 3d, confidence bands are not necessary for this differentiation.

SECOND ROUND COMMENT: OK

SECOND ROUND RESPONSE: Thanks

14. FIRST ROUND COMMENT: The use of the Kashchiev equation (Reference 27) for the calculation of the diffusion coefficient from the induction times is commented, but the equation is not shown.

FIRST ROUND RESPONSE: To address this comment, we have added the Kashchiev equation into the Supplementary Information file. See the section named as "Using the Kashchiev expression to calculate the diffusion coefficient from the induction time for the critical size at the nucleation temperature".

SECOND ROUND COMMENT: OK, thank you

SECOND ROUND RESPONSE: Thanks

15. FIRST ROUND COMMENT: The development temperature, T_d (1073 K), used for the some data points of the lowest temperature analyzed is different from the T_d (1119 K) used for the first data points and also for all the other temperatures. The time shift, t_s , is a function of T_d , which also affects the $t_{(ind_d)}$ that was determined from the linear adjustment of the last points and, consequently, the $t_{(ind_n)}$ determined by equation S-3. The authors should comment on how the use of these different T_d would affect their calculations?

FIRST ROUND RESPONSE: The Kashchiev treatment can only be used for the induction time for the critical cluster size at the nucleation temperature. That is why we corrected the measured induction times (which were for the critical cluster size at the growth temperature) using the expression developed by Shneidman and Weinberg.

As discussed on page 81, figure 11 of chapter 3 in “*Nucleation in Condensed Matter – Applications in Materials and Biology*,” K. F. Kelton and A. L. Greer, Elsevier (2010), this approach gives a valid correction to the data so that the standard Kashchiev treatment can then be used. Both the correction and the standard Kashchiev treatment have been embedded in equation S-1 in the Supplementary Information file. Details of the derivations of equation S-1 can be found in *Journal of Non-Crystalline Solids* 525 (2019) 119575.

The different growth temperatures have already been included in our calculations. Specifically, when we input the nucleation data at 948 K into equation S-1, we also input S-1 the related growth temperature, 1073 K. When we input the nucleation data at 998 K, 1011 K, and 1023 K into equation S-1, the related growth temperature, 1119 K is also input. The details about the iterative calculation steps can be found in the analysis steps part of the Supplementary Methods in the Supplementary Information file.

SECOND ROUND COMMENT: *OK, thank you.*

SECOND ROUND RESPONSE: Thanks

16. New minor comment from SECOND ROUND: The title is considerably long but more appropriate than the previous one since the test was performed only on one glass. Data for this particular Ba-silicate glass are available just in publications of the Kelton group.

SECOND ROUND RESPONSE: In the new manuscript, we have updated the title to meet the 15 words limit from the Journal. The title has been changed from “The Low-Temperature Anomaly in the Steady-State Nucleation Rate in Silicate Glasses is Shown to be Artifact in a 5BaO·8SiO₂ Glass” to “Low-Temperature Nucleation Anomaly in Silicate Glasses Shown to be Artifact in a 5BaO·8SiO₂ Glass”. It is true that currently we are the only group who have studied 5BaO·8SiO₂ glass and who have tested the low-temperature nucleation anomaly using 5BaO·8SiO₂ glass.

17. New minor comment from SECOND ROUND: Introduction: "A barium-silicate glass was chosen since they have higher nucleation rates than other glasses, such as lithium disilicate or soda-lime silicate so that it takes less time to obtain a significant number of nuclei.": Some soda-lime silicate glasses may have higher nucleation rates (e.g., the 2-1-3 glass) than barium silicates.

SECOND ROUND RESPONSE: Thank you for this point. To address this point, in this sentence, we have changed “lithium disilicate or soda-lime silicate” into “Li₂O□2SiO₂ or Na₂O□2CaO□3SiO₂”. Also, this sentence has been moved to the methods section.

18. *New minor comment from **SECOND ROUND**: Results, line 5: It would need to clarify that it refers to the induction time at the growth temperature (as written in line 7, page 6), since this is the first time this variable is mentioned, and this is not a specific glass science journal.*

SECOND ROUND RESPONSE: Thank you for raising this point. Saying that the induction time is the induction time at the growth temperature at the beginning of the Results section will not help present the results themselves and it will confuse the readers. The paragraph about the analysis procedures, where it is necessary to mention this nuance in the induction time to proceed with the analysis is a better place to introduce this subtlety.

19. *New minor comment from **SECOND ROUND**: Conclusion: "For the new measurements reported here the glasses were given a much longer nucleation treatment than was used in all previous measurements. These new data do not show a low-temperature anomaly": This paragraph is somewhat confusing; it implies that various nucleation rates at different temperatures were determined.*

SECOND ROUND RESPONSE: Thank you for this point. The usage of plural here refers to that there are measurements in different samples, i.e. 15 days sample, 35 days sample, 55 days sample, etc. We did not state that we determined various nucleation rates at different temperatures.

20. *New minor comment from **SECOND ROUND**: The used heating rates of the double stage heat treatment and their possible effects on IST were not reported.*

SECOND ROUND RESPONSE: Thank you for this point. Instead of ramping up the temperature of the furnace and the sample together by setting a thermal program of the furnace, we inserted the samples into the furnace after the furnace equilibrated at the nucleation or growth heat treatment temperature. In this method, the effect of the heating rate is negligible. Moreover, since the samples were heated for at least 15 days, the effect of the heating rate is again negligible.

21. *New minor comment from **SECOND ROUND**: Measurement of possible athermic nuclei were not reported.*

SECOND ROUND RESPONSE: Thank you for this comment. The number of nuclei in the as-quenched glass is negligible as $3.48 \times 10^{-5} \pm 1.25 \times 10^{-5} \mu\text{m}^{-3}$. As mentioned in the end of the Methods section, the density of nuclei in the as-quenched glass has already been applied in our calculations.

22. *New minor comment from **SECOND ROUND**: Overall this is a very good paper, with a few problems.*

SECOND ROUND RESPONSE: Thanks.

Response to Reviewer #3:

1. *New comment from SECOND ROUND: Owing to its huge fundamental and applied merit and the quality of the investigation, I certainly think that the manuscript by Xia et al. should be published in Nature Communications. I suggest a few mostly editorial revisions that will significantly enhance the impact of this paper.*

In the Introductions, please specify that previously measured nucleation rates below a certain temperature were significantly lower than the prediction of CNT based on assuming slower kinetics due to greater viscosity at the lower temperatures. This discrepancy enforced the assumption that the nucleation barrier shoots up. This explanation will tie the misinterpretation of the previous results with your powerful demonstration that the nucleation rates measured before steady state is reached are lower.

SECOND ROUND RESPONSE: Thank you for your comments in the First Round for *Nature Materials* and in the Second Round for *Nature Communications*. Since we didn't see any further comment from you upon our response to your First Round comments, here we are answering only the new comments from you in the Second Round.

In the introduction section, we have already mentioned the problems with the previously measured nucleation rates and analysis. While it could appear that the larger viscosity below T_g could be the source of the departure from the predictions of the CNT, that is not the case. The viscosity is measured independently from the nucleation measurements and the extrapolated values from the supercooled liquid are typically used. Since the functional form of the viscosity is not known and since structural relaxation effects may be important, we prefer to use a more direct measure of the kinetics, the induction time for nucleation. The induction time will reflect the same kinetics governing the nucleation rate. We feel that including a discussion of the viscosity could be misleading and prefer not to do that.

2. *New comment from SECOND ROUND: Along the same line of thought, please use Ref. 28 or the more recent book by Kashchiev to explain why nucleation rates measured before steady state is reached are slower and not faster than the steady state values; this transition imposes a superlinear correlation $N(t)$, so nicely demonstrated in Fig. 2. p. 4: please denote the increase of the number of crystals*

that nucleate per unit volume per unit time as superlinear instead of the more generic nonlinear. Do not use parentheses to bracket the values of I_{st} and θ . Please report the values of I_{st} and θ with fewer significant digits as demanded by the standard deviations of the respective measurements. Consult a book on statistics of experiment if needed.

SECOND ROUND RESPONSE: Thank you for this comment. It is true that under particular situations the nucleation rate can be larger than the steady state rate. This can happen, for example, when the glass is given several specific heating treatments prior to the nucleation treatment (see *Nucleation in Condensed Matter*, K. F. Kelton and A. L. Greer, Elsevier (2010), pages 312-315). However, the glasses that we studied were directly quenched from the liquid. The cluster population is, then, smaller as a function of size than the steady-state population at the nucleating temperature. Since to lowest order the nucleation rate is proportional to the cluster density at the critical size multiplied by the forward rate constant, the nucleation rate must be smaller than the steady-state value. All experimental measurements and analytical treatments support this. Since this is widely recognized in the glass community, we do not feel that it should be discussed in explicit detail here. To address this point, in the related sentence in the first paragraph of the Results section, we have added “a phenomenon widely recognized for nucleation in melt-quenched glasses as due to the evolving cluster population as a function of cluster size”. Regarding the second point, we have deleted the parentheses bracketing the values of I_{st} and θ . We have rounded the standard errors of I_{st} and θ to one significant figure, and the I_{st} and θ have been rounded to the same digit accordingly. “ 397 ± 18 ” is rounded to “ 400 ± 20 ”. And “ 39638.5 ± 3229.3 ” is rounded to “ 40000 ± 3000 ”. In Table 1, for the I_{st} and θ values from the previously published paper, we have kept the values as they are written in the previous literature without further rounding.

3. *New comment from SECOND ROUND: Please use same units of volume and time throughout. Volume is now reported in cubic microns and cubic millimeters, whereas time is measured seconds, minutes and even days. This makes verifying the stated parameter values with the presented data confusing.*

SECOND ROUND RESPONSE: In Figure 2, the data are plotted as a function of the number of days and the number of nuclei per cubic microns is useful to give the reader a better intuition about the experimental time and results. Also, cubic microns is better linked to the length scale in the SEM images. In Table 1, we report the nucleation rate values in per cubic millimeters per second and induction time values in per minutes because we want to compare the values obtained in this measurement with previous values in the literature. The mixture of number per cubic millimeters per second and per minutes to report nucleation rate and induction time is common in studies of nucleation in the silicate glass field (see *Journal of Non-Crystalline Solids* 163 (1993) 1-12, and

Journal of Non-Crystalline Solids 74 (1985) 373-394, as examples). We have followed this standard.

4. *New comment from SECOND ROUND: In Fig. 3b, the legends that state growth T 1093K and growth T 1119K in both black and red are confusing. It is not clear if the points belong to the figure legend or to the linear correlation starting at lower temperatures.*

SECOND ROUND RESPONSE: Thank you for this comment. We appreciate the referee's concern but unfortunately all the information in Fig. 3b is necessary. We do not see a way to make it simpler and visually clearer. In Fig. 3b, each of the terms "growth T 1073K" and "growth T 1119K" is a description about the term's nearest data point in the figure. Those three points are real data, which are used in the nucleation data analysis procedure. After reading the main legend, i.e. the symbols and colors of "this work", "linear fitting", "previous work", and the axes, we feel that the information regarding these growth temperature points can be understood.